# Transcriptomic and Physiological Response of Durum Wheat Grain to Short-Term Heat Stress during Early Grain Filling

**DOI:** 10.3390/plants11010059

**Published:** 2021-12-25

**Authors:** Anita Arenas-M, Francisca M. Castillo, Diego Godoy, Javier Canales, Daniel F. Calderini

**Affiliations:** 1Institute of Biochemistry and Microbiology, Faculty of Sciences, Universidad Austral de Chile, Valdivia 5110566, Chile; anitamaribel@gmail.com (A.A.-M.); castillo.francisca88@gmail.com (F.M.C.); 2ANID—Millennium Science Initiative Program-Millennium Institute for Integrative Biology (iBio), Santiago 8331150, Chile; 3Plant Production and Plant Protection Institute, Faculty of Agricultural Sciences, Universidad Austral de Chile, Valdivia 5110566, Chile; Diegoe.Godoyk@outlook.com

**Keywords:** durum wheat, DOF transcription factor, grain weight, grain quality, gene regulatory network, heat stress, RNA-seq

## Abstract

In a changing climate, extreme weather events such as heatwaves will be more frequent and could affect grain weight and the quality of crops such as wheat, one of the most significant crops in terms of global food security. In this work, we characterized the response of *Triticum turgidum* L. spp. *durum* wheat to short-term heat stress (HS) treatment at transcriptomic and physiological levels during early grain filling in glasshouse experiments. We found a significant reduction in grain weight (23.9%) and grain dimensions from HS treatment. Grain quality was also affected, showing a decrease in starch content (20.8%), in addition to increments in grain protein levels (14.6%), with respect to the control condition. Moreover, RNA-seq analysis of durum wheat grains allowed us to identify 1590 differentially expressed genes related to photosynthesis, response to heat, and carbohydrate metabolic process. A gene regulatory network analysis of HS-responsive genes uncovered novel transcription factors (TFs) controlling the expression of genes involved in abiotic stress response and grain quality, such as a member of the DOF family predicted to regulate glycogen and starch biosynthetic processes in response to HS in grains. In summary, our results provide new insights into the extensive transcriptome reprogramming that occurs during short-term HS in durum wheat grains.

## 1. Introduction

The major challenges that global agriculture is facing are the rise in food demand from the increasing population and climate change. In this context, crop yields should increase ~60% by 2050 to maintain the global food supply, but at the same time, a reduction in the global cultivated area is expected because of the decrease in rainfall and the temperature increase [1,2,3]. On the other hand, greenhouse gas emissions from human activities will raise the global surface temperature, which will probably increase between 1 and 5.7 °C by 2081–2100 [4]. The increase in the frequency of extreme weather events such as heatwaves and droughts has also been forecasted at a global scale, as it has already been recorded since the 1950s [4].

Heat waves will negatively impact different cropping systems as high temperatures for short periods have the potential to significantly reduce staple food crop production and affect its quality [5]. Heat-stress (HS) episodes affect plant and crop physiology in several ways [6], especially affecting edible organs. The effects of HS on plants and crops depend on (i) the temperature increase, (ii) the duration of the increased temperature, and (iii) the plant/crop developmental stage [7]. The most heat-susceptible phase in crops is the reproductive period [8,9,10].

In cereals like wheat, HS causes a significant reduction in grain yield components and quality traits under moderately high temperature [8,9,10,11], while heat shock temperatures can reduce grain number (sterility and abortion of grains) during anthesis to early grain filling (GF), and at the same time decrease GF [12]. These studies have suggested that the wheat sensitivity to HS largely depends on the timing and intensity of the stress. The impact of higher temperature is particularly risky between booting and the beginning of GF [13,14,15]. A key sensitive period of wheat-growing grains is when cellularization of the endosperm is set (i.e., ~7–10 days after anthesis (DAA)) and at the start of the high-rate accumulation of starch and proteins during the early GF (i.e., 10–15 DAA) [8,16,17]. Supporting these key moments, recently Herrera and Calderini (2020) [18] showed a close association between final grain weight and the maximum pericarp dry weight in wheat, which is reached between 110 and 235 °C after anthesis (around 10 DAA). These results highlight the importance of the early GF phase in addition to previous reports accounting for grain length and water content stabilization, which are discussed later [19,20,21].

The most common forms of domesticated wheat are tetraploid durum wheat (*Triticum turgidum* L. ssp. *durum* (2n = 4x = 28, AABB genome)) and hexaploid bread wheat (*Triticum aestivum* L. (2n = 6x = 42, AABBDD genome)). These domesticated kinds of wheat are derived from interspecific hybridization events, showing a positive correlation between increased ploidy and the ability to grow and yield [22]. Molecular mechanisms underlying the response to short-term HS have been widely reported for the hexaploid wheat, focusing on seedlings, flag leaf [23,24,25], or grains [26,27,28,29], but the regulatory stress mechanisms in tetraploid durum wheat are still poorly understood. One of the first works that characterized the molecular mechanisms of durum wheat in response to HS was reported by Aprile et al. (2013) [30], who performed a transcriptomic analysis of the flag leaf subjected to short HS at the booting stage using microarray technology. This study showed a remarkable induction of genes encoding for the enzymes involved in the fatty acids β-oxidation and the glyoxylate cycle.

The recent report accounting for the sequencing and assembly of three tetraploid wheat genomes [31,32] represents an outstanding opportunity to improve the knowledge of molecular mechanisms associated with both the sensitivity and the resilience of durum wheat under HS. The quality of the genome assembly of modern durum wheat is comparable to the RefSeq v1.0 annotation of bread wheat [33], which allows the study of gene function in durum wheat for agronomic traits [32].

In addition, the transcriptomic response of the flag leaf after 24 h of HS in Australian durum wheat was published, showing that differentially expressed genes (DEGs) were associated with hormone signaling, photosynthesis, and metabolic processes [34]. Moreover, this transcriptome profiling discovered many positive stress regulators that are activated, such as reactive oxygen species (ROS) scavenging enzymes, chaperone proteins, and protein phosphatases, to either alleviate cellular damage or adjust biological processes, leading to adaptive physiological changes [34]. However, the initial transcriptomic response of grains to HS during early GF has not yet been analyzed in durum wheat. Moreover, the molecular factors controlling the early events of the HS response of durum wheat grains remain unknown. Therefore, the knowledge of the adaptive responses to HS is the focus of current research to contribute to crop adaptation in the future.

In this context, a relevant question is to what extent the thermotolerance mechanisms described so far for the flag leaf or seedlings explain the final grain yield of durum wheat under HS conditions. It is important to note that the grain yield of wheat is not limited by the source of photosynthates, and the GF is a key period for yield and quality performance [35]. At the molecular level, studies about the transcriptomic response of grains in durum wheat are still scarce. These kinds of works can help to identify specific new candidate genes associated with complex mechanisms and traits such as thermotolerance, yield, and grain quality [36]. In parallel, over the last decades, there have been major advances in the gene regulatory network (GRN) prediction methods that aim to map all transcription factors (TFs) and their target gene interactions from genome-scale data sets [37].

In this study, we present the physiological and transcriptomic response of durum wheat grains to HS during early GF to identify key heat-stress response genes and their predicted regulatory networks. We found that a short-term HS during early GF significantly affects grain weight and quality, and the initial response of grain to HS is characterized by the induction of chaperones, together with the inhibition of genes related to proteolysis and transcription regulation. Furthermore, a TF-target network analysis allowed us to identify a novel TF of the DOF family as a key regulatory factor in biological processes associated with grain quality in durum wheat. Finally, our results can be useful in forming an integrated strategy to develop heat-stress-resilient crops in the future. Improving crop resilience to HS has become a major target for breeding programs under the global climate change constraint.

## 2. Materials and Methods

### 2.1. Plant Material and Thermal Treatments

Seeds from *Triticum turgidum* spp. *durum* genotype Queule-INIA were sown at a density of three seeds per pot and fertilized with 150 kg/ha of nitrogen (N) and 300 kg/ha P_2_O_5_. In addition, 150 kg/ha N was applied to average tillering. Plants were grown in ambient conditions from sowing until close to the heading stage. Then the pots were moved inside the glasshouse to continue the plant’s development. Pots were periodically surface-irrigated to avoid water shortages. The crop phenology was followed every 5 days starting at emergence, using the decimal code scale [38]. The timing of physiological maturity was estimated when grains of wheat reached a water concentration of 37% [39] (data not shown). At anthesis (Zadok 65), principal spikes with similar development and size were tagged. Two thermal treatments were performed: control treatment (to ambient glasshouse temperature) and HS, which consisted of increasing the temperature in a range of moderately high temperatures (20 to 32 °C). To achieve this, the temperature inside the HS chambers was increased 10 to 15 °C on average above the control temperature. The thermal treatments were set at the early GF phase, specifically at 10 until 14 DAA, when the effect of elevated temperatures on grain number per spike was negligible [40], and the other yield components such as weight and quality were beginning to set in developing seed. To perform the thermal increase, HS chambers of transparent polyethylene (0.15 microns) were built (Figure 1a). Each chamber (1.2 × 1.0 × 1.2 m) was equipped with thermostatically controlled electric heaters as in previous experiments [8,41]. The thermal regime was controlled by thermal sensors placed at the spike height that recorded the air temperature inside the HS chambers and the ambient air temperature of the glasshouse and connected to a temperature regulator (Cavadevices, Buenos Aires, Argentina). Five independent experiments for thermal treatments were performed with groups of plants with the same phenology. For a selected spike, the temperature inside the grain was recorded during the HS treatments (Figure 1a, middle and right panels). The air temperature was recorded using data loggers (Extech Instruments SDL200 FLIR System, Inc, Nashua, NH, USA). Additionally, grain temperature was measured with a Teflon thermocouple (0.25 mm diameter, Extech Instruments). The thermocouple was set in the basal grain of the central spikelet from one spike in five independent experiments.

### 2.2. Plant Sampling and Physiological Measurements

After physiological maturity, only principal spikes (from at least 12 plants per replicate) from both thermal treatments were harvested, and all-grain positions pooled. Grain dimensions (length, width, and height) were recorded immediately after sampling the spikes using an electronic caliper (6-inch 150 mm Digital Calipers, Shanghai, China). Then samples were dried at 65 °C for 48 h in a heater (Binder Model FED-720, Tuttlingen, Germany). After that, pools with 50 seeds were grouped (3 to 4 pools per replicate) and weighed in an electronic balance (Mettler, Giessen, Germany) to obtain the dry weight. Individual grain weight was measured in an electronic balance from 1 to 2 pools per replicate. For each thermal treatment, three biological replicates were analyzed.

### 2.3. Grain Starch and Protein Concentration

Total grain starch was quantified from the main spikes (at least 12 spikes per replicate). Pooled samples were then generated from all-grain positions of the spikes. Grains were dried at 65 °C for 48 h; afterward, those grains were ground in a mill (Perten Instruments, Springfield, IL, USA) to obtain wheat flour. A total of 10 mg of wheat flour (for each sample) was used to measure total starch using the Starch Colorimetric/Fluorometric Assay Kit (BioVision, Inc., San Francisco, CA, USA), with minor modifications. The kit is based on the hydrolysis of the starch to glucose, which is oxidized to generate color. Absorbance was read at 570 nm in a Nanoquant Infinite M200 spectrophotometer (Tecan, Männedorf, Switzerland). Nitrogen (N) concentration of grains was measured in the wheat flour from the grain pools, as previously described. The Kjeldahl procedure was used to determine total N. Grain protein was calculated by multiplying the total N by 5.7 [42]. For each thermal treatment, three biological replicates were analyzed.

### 2.4. RNA Isolation and Quantitative Real-Time PCR Analysis

Within 3 to 12 h of starting HS, principal spikes were collected from 6 plants (3 plants per pot). Grain samples (complete caryopsis) were pooled from two basal grains of four central spikelets of each spike and quickly frozen in liquid nitrogen until processed. Three biological replicates for each timing and thermal treatment were performed. The RNA extraction protocol was based on [43] with modifications. In an Eppendorf tube 700 μL of extraction buffer (Tris 100 mM pH 8.0, EDTA 30 mM, NaCl 2M, and CTAB 3%, previously sterilized), 2% PVP40, 2% PVPP, and 4% 2-b-mercaptoethanol were mixed and pre-heated at 65 °C. A total of 50–100 mg of ground and frozen grain samples was added to the pre-heated extraction buffer. The mixture was then incubated at 65 °C for 2 min. A total of 650 μL of chloroform/3-metil-1-butanol (24:1) was added and mixed vigorously. This was centrifuged at 11,000 g for 5 min at 4 °C. The upper aqueous phase was recuperated and ½ volume of ethanol absolute was added and mixed. Then 600 μL of the sample was loaded in a NucleoSpin^®^ Gel and PCR Clean-up column (Macherey Nagel) and centrifuged at 11,000 g for 3 min at 4 °C. Following this step, 600 μL of wash buffer was added and centrifuged at 11,000 g for 3 min at 4 °C. Flow-through was discarded and the column was dried by repeating the centrifugation step for 10 min. The column was placed in a new tube, and 40 μL of nuclease-free water was added and incubated at room temperature for 5 min. After that, the column was centrifuged at 11,000 g for 5 min at 4 °C and the eluted RNA was kept on ice. DNase reaction RQ1 RNase-Free DNase (Promega, Madison, WI, USA) was prepared following the manufacturer’s instructions. The whole volume of eluted RNA was incubated at 37 °C for 30 min. After that, 350 μL of NTI binding buffer and 200 μL of ethanol absolute were mixed with the DNase reaction and loaded into a new PCR Clean-up column, then centrifuged at 13,000 rpm for 3 min at 4 °C. The flow-through was discarded and the step of drying the column was repeated. Finally, the RNA was eluted by adding 35 μL of nuclease-free water into the column and incubating at room temperature for 5 min. After that, it was centrifuged at 13,000 rpm for 5 min at 4 °C. The eluted RNA was kept at –20 °C for future analysis. A total of 500 ng of RNA was used to prepare first-strand cDNA using the 5X All-In-One RT MasterMix (Applied Biological Materials Inc. (abm), Vancouver, Canada) according to manufacture instructions. Gene expression was measured by Touchdown qPCR assays (Zhang et al., 2015) with Brilliant II SYBR Green QPCR Master Mix (Agilent Technologies, Inc., Santa Clara, CA, USA) and an AriaMx Real-Time PCR System (Agilent Technologies, Inc., Santa Clara, CA, USA). The raw fluorescence data were analyzed with Real-time PCR Miner software [44]. Each qPCR sample (25 μL total volume) contained 25 ng of cDNA and each primer was 200 μM. The following PCR program was used: one cycle at 95 °C for 10 min; three cycles at 95 °C for 20 s, followed by 66 °C for 10 s, during which the temperature decreased by 3 °C per cycle; and 40 cycles of 95 °C for 20 s, 55°C for 10 s, and 72 °C for 10 s. The gene-specific forward and reverse primers used are listed in Appendix A. The ortholog gene of TraesCS4A02G414200 (a putative ubiquitin-conjugate enzyme; TRITD4Av1G226680), previously described with stable expression across the development of seeds in wheat [45], was used as an internal control to quantify the relative mRNA levels.

### 2.5. RNA-Seq Analysis

Spikes from the main spike strata were harvested 3 h after thermal treatments (control and HS) from 10 to 12 plants for transcriptome analysis. Basal grains (G1 and G2) from central spikelets of the spikes were sampled and immediately frozen in liquid nitrogen. Three biological replicates were analyzed for each treatment. Library construction and sequencing were carried out by the sequencing service unit of the Faculty of Sciences in the Pontifical Catholic University of Chile (Santiago, Chile). Sequenced reads were pseudo-aligned to the publicly available Triticum turgidum transcriptome obtained from Ensembl Plants using Kallisto (v0.46.1) [46]. The transcript indices for Kallisto were generated from *Triticum turgidum* annotation version Svevo.v1, which includes 196,153 cDNAs (http://plants.ensembl.org/Triticum_turgidum, 25 April 2021). Differentially expressed genes were determined using the Wald test implemented in the R package Sleuth [47] with a *q-*value < 0.05 and an absolute fold of change (FC) > 1.5. The data were deposited in NCBI’s Gene Expression Omnibus (Edgar et al., 2002) and are accessible through GEO Series accession number GSE186472 (https://www.ncbi.nlm.nih.gov/geo/query/acc.cgi?acc=GSE186472, 26 October 2021). The TF-target predictions obtained from the wheat GENIE3 network [48] were used for the construction of gene regulatory networks considering the Ensembl Plants orthology annotation between *Triticum aestivum* and *Triticum turgidum* (http://plants.ensembl.org/, 26 October 2021). Only TF-target predictions with a connectivity higher than 0.005 in the wheat GENIE3 network [49] and significant regulation by heat stress (*q*-value <0.05 and absolute fold of change (FC) > 1.5) were considered for the network construction. The resulting gene regulatory networks were visualized in Cytoscape v3.8.2. In addition, we used the IVI algorithm for the identification of the most influential nodes from each network implemented in the “influential” R package [50].

### 2.6. Gene Ontology (GO) Enrichment Analysis

Gene functional enrichment analysis was performed using the gprofiler2 R package with Benjamini–Hochberg multiple testing correction [51]. Then, we summarized the enriched term lists using the REVIGO online ontology analysis tool to reduce the redundancy of GO terms [52]. Finally, we selected the top 5 most enriched and non-redundant GO terms of the biological process domain as well as the most enriched GO term of the cellular component and molecular function domains in the Manhattan plots.

### 2.7. Transcription Factor Binding Site (TFBS) Enrichment Analysis

Position weight matrices of TF binding sites were obtained from the JASPAR 2020 database [53]. CiiiDER software was used to predict TF binding sites across the first 500 bp upstream regions from the predicted transcriptional start site of the 432 targets of TRITD5Bv1G096580 (a Zinc finger DOF-type) using a site identification deficit threshold of 0.15 [54]. Genes with a low significance in response to heat stress (*q*-value > 0.1) were included in a background promoter sequence set. Then, the enrichment of TFBSs was assessed comparing the distribution of TF binding sites predicted in the foreground as well as the background promoter sequence set (*q-*value threshold of 0.01).

## 3. Results and Discussion

### 3.1. Environmental Conditions and Temperature Increment

One hundred pots with three seeds of tetraploid durum wheat each were sown at the end of January 2016 and grew under ambient open-door condition until the end of March 2016, when daily mean temperatures dropped below 20 °C in the Experimental Station of the Universidad Austral de Chile in Valdivia (39°47′ S, 73°14′ W, 19 m asl), Chile. Pots were moved into a glasshouse facility to continue plant development and the set of thermal treatments. Across the plant phenology, the mean of air temperatures was stable: daily mean temperature from emergence (Zadok 09) to anthesis (Zadok 65) was 16.5 ± 2.2 °C and from anthesis to physiological maturity (Zadok 89) was 16.7 ± 2.6 °C (Appendix A).

Five independent experiments for thermal treatments were performed with groups of plants with the same phenology. At anthesis, principal spikes were labeled. Only spikes similar in size and phenology from the main stems were considered for both control and thermal treatment measurements. In each experiment, eight pots, each with three plants, were introduced into the HS chambers and the same number of pots were left outside the chambers as control. Plants at 10 DAA in polyethylene chambers (Figure 1) were set to undergo the HS treatments for 4 days. The average ambient air temperature (control condition) was 14.8 ± 5.5 °C, while the mean of air temperature inside the HS chamber was 26.3 ± 5.2 °C during the 4 days of HS through five independent experiments (Figure 1b,c). Grain temperature was measured during the HS treatments, which averaged 23.6 ± 4.0 °C (Figure 1c). This meant that under the HS condition, the temperature increments reached 11.5 °C on average, but grain temperature increased 8.8 °C on average with respect to the control condition. According to these results, the difference between the air and grain temperatures inside the HS chambers was 2.7 °C on average.

Temperatures > 30 °C represented a high-temperature threshold, causing significant grain damage [55], especially when the period in which this temperature threshold was exceeded was greater than 90 h during grain filling [56]. In our experiments, during the 4-day period of thermal treatments, plants experienced maximum temperatures >30 °C for 17.2 ± 8.3 and 1.4 ± 3.5 h on average under HS and control conditions, respectively, across the five experiments (Appendix A). In addition, important differences were found between air and grain tissue temperatures in the literature. For example, the temperature of the spikelet tissue in rice was lower than the ambient air temperature by 0.4–1.8 °C when air temperature ranged between 30 and 38 °C, respectively [57], which is lower than the difference between air and grain temperature inside the HS chambers found in our experiment (2.7 °C). However, in irrigated wheat experiments carried out under field conditions, spike temperature was lower than ambient temperature by 5 °C through the grain filling [55], but under rainfed conditions, spike temperature showed a lower temperature than the air only during the early phase of GF. During the middle GF (between 15 and 18 DAA), the spike temperature was only 0.3 °C lower than the ambient temperature, and from 28 DAA until physiological maturity, the spike temperature exceeded the ambient by up to 1.5 °C [55]. Recently, a small but significant difference of 0.29 °C was reported between the grain temperature of wheat and the ambient air temperature in a pot experiment [35]. On the other hand, the higher temperature measured in plant organs over the air was reported for the sun-flower capitulum, where its temperature was higher than the environmental air temperature by up to 8 °C [58]. The differences could be related to the stomatal conductance of various organs contributing differently to the transpiration in the course of capitulum maturation [58] and the differential exposures of plant tissues to sunlight.

### 3.2. Seed Size, Grain Weight, and Grain Quality Were Negatively Affected by Short HS at Early Grain Filling

Grain weight (GW) was highly affected (*p* < 0.0001) by HS, decreasing 23.9% on average relative to control (Figure 2a). Grain dimensions showed a high, although a little lower sensitivity, to HS by each dimension with a decrease of 11.7% (*p* < 0.001), 14.4% (*p* < 0.0001), and 14.0% *(p* < 0.01) in grain length, width, and height, respectively (Figure 2b). The impact on both grain weight and grain size can be seen in Figure 2c. A wide range of GW categories at harvest, i.e., from 10 to 75 mg, was recorded between thermal treatments (Figure 2d). In addition, heat-stress treatment modified the GW category distribution by moving them from heavier to lighter values compared to the control (Figure 2d).

Grain starch in the control treatment reached 74.6% of dry weight (DW), while grains from the HS treatment only reached 59.1% DW; therefore, the grain starch was affected by HS decreasing 20.8% with respect to control (*p* < 0.05) (Figure 2e). Previous studies have reported a reduction in the activity of soluble starch synthase in wheat grains at high temperatures [59] and under HS conditions [60], providing evidence that under thermal stress these enzymatic factors are relevant when defining the final impact on grain yield. The opposite effect was observed in protein content, where grains from HS treatments had an increment of 14.6% (*p* < 0.001) with respect to the control (Figure 2f). In agreement with our results, several works reported positive HS impacts on grain protein concentration, most of them on *T. aestivum* [29,61,62,63,64,65,66]. However, other studies have shown a small or nonexistent effect of increased temperature on the grain protein concentration of hexaploid wheat under moderately high temperatures during most of the grain filling period [8,67]. These differences could provide evidence of the wide genetic variability among wheat genotypes in this response or differences in methodology and manipulations among treatments. In a recent study, the increase in protein content in mature heat-stressed grains suggested that more energy and greater assimilation were allocated to the deposition of protein reserves, even though the metabolomic and proteomic data indicated an overall decrease in proteins involved in metabolic and protein-synthesis processes under HS [66]. Thus, HS events further showed a significant negative effect on grain dry mass, but at the same time could have a positive effect on protein concentration in flour as was shown by many studies [65]. Likely, this is due to the differential effect of HS on either carbon or nitrogen balance of grains and their timing of upload and synthesis in grain endosperm cells [68]. Additionally, it has been stated that the protein quality is highly influenced by the frequency of high temperatures during GF and this trait declines with exposure to elevated temperatures over a long time [56].

### 3.3. Temporal Expression Profiles of HS Marker Genes in Durum Wheat Grains

To determine the dynamics of gene expression during the first hours of HS treatment, we evaluated the relative expression of several HS gene markers previously described at transcript level as highly upregulated under heat conditions in grains of hexaploidy wheat [26]. Heat shock proteins (HSPs) are crucial for tolerance to high temperatures in plants acting as molecular chaperones to promote the correct folding and counteract aggregation of proteins under HS [69,70,71]. In the case of HSPs, we selected HSP101 and HSP23.5 for expression analysis by qPCR (Figure 3), which belongs to the HSP100 and small HSPs families respectively. Members of both families have been shown that play a crucial role in thermotolerance acting as molecular chaperones protecting their targets from denaturation and aggregation [72,73,74]. In addition, the Rubisco activase, an essential enzyme for photosynthesis that removes inhibitory sugar phosphates from the active sites of Rubisco, was shown to be inducible by heat and was consistent with its role in maintaining Rubisco integrity at high temperatures [26,75].

We found significant differences (*p* < 0.01) in the relative expression of three HS gene markers between HS samples and the control treatment at 3, 6, and 9 h after starting the thermal treatments (Figure 3). The gene expression profiles of the HS gene markers in grain showed a strong induction in response to HS during the first hours of thermal treatment, and within 10 h of all HS markers evidenced a significant reduction in their expression (Figure 3). Interestingly, the maximum induction was reached 6 h after the start of the HS treatment. This result suggested that transcription factors (TFs) that mediate the initial heat-stress responses (HSRs) should be induced early [76,77]. On the other hand, the main regulatory factors controlling the spatial and temporal expression of the HSPs are the plant heat-stress transcription factors (HSFs). Thus, the HSFs, HSPs, and specific *cis*-elements form a regulatory circuit that helps in the transcriptional activation to let HSR genes ultimately establish tolerance of heat stress [78].

One of our objectives was to identify transcription factors involved in the initial response of durum wheat grains to heat stress as well as their putative target genes, associated with key biological processes for grain yield and quality. Thus, to identify primary TFs mediating HS-response in wheat grains, we performed a transcriptomic analysis on grains at 3 h of initiating the HS treatment.

### 3.4. The Initial Transcriptomic Response to HS in Grain Is Characterized by the Induction of Chaperones Together with the Inhibition of Related Genes to Proteolysis and Transcriptional Regulation

To gain new insights into the molecular factors underlying the heat-stress response of durum wheat grains, we performed a transcriptomic analysis of grain samples obtained from durum wheat subjected to HS during early GF. Total RNA was extracted from the complete grain at 10 DAA 3 h after initiating the heat stress with three replicates for control and heat treatment. The RNA-Seq data were pseudo-aligned to the *Triticum turgidum* transcriptome (annotation version Svevo.v1) using Kallisto [46]. On average, each sample had 29.4 M reads, of which 20.7 M mapped (70.4%) (Appendix A). We found that 41.698 genes were expressed using the default filtering function of the Sleuth software, which required at least five mapped reads for a transcript in 47% of the samples [47]. Then, to identify DEGs we used as criteria an absolute 1.5-fold change (FC) and a *q*-value < 0.05. Using these criteria, 1590 DEGs were identified between control and HS (Figure 4a); of them, 671 DEGs belonged to induced genes and 919 DEGs were repressed genes (Figure 4a) (Appendix A). In the top ten downregulated genes (in order of statistical significance) (Figure 4b), the genes TRITD6Bv1G047750, TRITD2Av1G291390, and TRITD7Av1G261660 related to peptidase activities showed one of the most prominent reductions in their expression (log2 FC: −1.87 to −5.42), suggesting that functions related to “proteolysis” are strongly inhibited in response to short HS in grain. On the other hand, a putative bZIP transcription factor (TRITD5Av1G026510), was highly downregulated (log2 FC −1.49) (Figure 4b). The bZIP proteins have been documented to play a role in hormonal responses, light signaling and photomorphogenesis, seed germination, maturation, floral induction, and flower development [79], and an ortholog of this gene “bZIP10” (AT4G02640) in Arabidopsis was implicated in the transcriptional activation of HSP90 in response to an increment in the glutathione levels in the leaves of stressed plants [80].

Genes encoding HSP and co-chaperones were in the top ten most significant upregulated genes (*q*-value < 9.1 × 10^−71^) (Figure 4b). With respect to the HSP, five annotated genes as “Alpha crystallin/Hsp20 domain” (TRITD3Bv1G044280, TRITD4Bv1G129640, TRITD3Bv1G044290, TRITD5Av1G157030, and TRITD3Bv1G012590) and two “Heat shock protein Hsp90 family” genes (TRITD2Bv1G010840 and TRITD0Uv1G009860) were induced with a log_2_ FC ranging from 1.43 to 3.06 (Figure 4b). The subfamily of small HSP in wheat is characterized by at least one α-crystallin domain that acts as the signature motif binding non-native proteins and is among the first to respond during stress conditions and the seed development stage [81]. On the other hand, HSP90 has been described as a key component of a negative regulatory mechanism of HSFs during the initial HS response. Specifically, HSP90 together with HSP70 directly interact with HSFA1 under optimal temperature, inhibiting its ability to interact with the transcriptional machinery, but upon heat stress denatured proteins deplete HSP90 from the HSP–HSF complex, which frees the transcriptional activity of HSFA1 [82,83]. Previously, other RNA-seq analyses of grains of bread wheat during HS at 1 and 4 h also reported the transcriptional response of HSP90 [27].

### 3.5. Gene Ontology (GOs) Analysis of DEGs under Short Heat Stress in Grains of Durum Wheat

To provide an overview of the biological pathways related to the up- and downregulated genes, we performed a gene ontology (GO) analysis using gprofiler2 software [51]. As expected, the biological process “response to heat stress” was one of the most enriched GO terms (Figure 5a). In fact, qPCR analysis of genes encoding for HSPs and co-chaperones showed that these genes were significantly induced by heat stress in grains at 3 h after treatment (Appendix A). Moreover, we found that photosynthesis, phenylpropanoid biosynthetic process, and erythrose 4-phosphate/phosphoenolpyruvate metabolic process were among the most enriched biological processes in upregulated genes (Figure 5a–c, Appendix A). In the case of the molecular function domain, the most significant GO terms were “carbon-nitrogen lyase activity” and “oxidoreductase activity” (Appendix A), while “photosystem I” (Figure 5c, Appendix A) was the most significant GO term of the cellular component domain. These results suggested that heat stress induces the expression of genes associated with photosynthesis and photosynthetic electron transport system, which was in agreement with previous work on bread wheat [26,75,84] and also Arabidopsis [85], and it has been suggested that this regulation may induce acclimation of photosynthetic CO_2_ fixation and photoprotection during short-term high-temperature stress [85].

In the case of downregulated genes, the top five enriched GO terms of the biological process domain were carbohydrate metabolic process, transcription, regulation of the metabolic process, and positive regulation of transcription (Figure 5b,c), suggesting that heat stress reduces the expression of genes related to carbohydrate metabolism and regulation of transcription and signal transduction. This inhibitory effect on the gene expression involved in carbohydrate metabolism likely causes a general decrease in seed carbohydrate content. In fact, a global decrease in the content of carbohydrate metabolites was previously reported in wheat filling grains at 15 DAA subjected to 3 days of HS [67]. Regarding the regulation of carbohydrate metabolism in storage organs, it has been shown that several members of the NAC TF family can directly regulate genes involved in starch biosynthesis in maize [86,87] and wheat [88]. Interestingly, the NAC-type TFs were the most abundant family between downregulated genes (13 out of 83 downregulated TFs) in our HS experiment, suggesting that this TF family is important during the heat-stress response in grains and might be related to the downregulation of carbon metabolism genes. In the same way, it was demonstrated in hexaploid wheat that TaNAC019 binds to the promoter of starch metabolism genes, and knock-out mutants of all three TaNAC019 homeologs exhibited reduced expression of key starch biosynthetic genes and significantly lower starch content than wild-type grains [88]. Interestingly, we found an ortholog of TaNAC019 in tetraploid wheat (TRITD3Av1G022150), which was significantly downregulated by heat stress in our experimental conditions (log_2_ FC 2.2 and *q*-value < 0.05), suggesting that this NAC TF is also involved in the regulation of starch metabolism genes in durum wheat grains in response to heat stress.

### 3.6. Identification of Regulatory Factors Associated with the Response to HS in Grains of Durum Wheat

The response to HS is tightly regulated at the transcriptional level in plants, TFs being the key players in regulating this stress response [89]. Gene regulatory networks (GRNs) are used to represent specific interactions of regulators of gene expression such as TFs with the expression of target genes [90,91]. To identify candidate TFs controlling HS response in durum wheat grains, we constructed two gene regulatory networks, one for genes upregulated and the other for genes downregulated by HS (Figure 6a). The TF-target predictions between DEG genes were obtained from wheat GENIE3 [48], which uses a machine-learning approach to predict the strength of putative regulatory links between target genes and their putative regulators using gene expression data [92]. In the case of genes upregulated by HS, we obtained a GRN with 63 TFs and 548 targets (Figure 6a; Appendix A). To identify the most relevant nodes in the GRN, we applied the IVI algorithm, which combines the most important topological characteristics of the network to rank the most influential TFs of the GRN [50]. In this manner, we identified seven members of the mTERF family as the most relevant nodes of this network (Figure 6a). Plant mTERFs are nuclear-encoded proteins capable of binding nucleic acids and regulating organellar gene expression and therefore are located in chloroplasts and/or mitochondria [93]. Several studies based on the phenotype of mTERF mutants suggested that these genes are essential for functional acclimation to diverse abiotic stresses, including heat stress [94]. For instance, it has been shown that an insertional mutant of mTERF18 is more heat-tolerant than the wild-type plants and exhibited higher transcript levels of HSPs in Arabidopsis [94,95].

In the case of downregulated genes, we obtained a GRN with 136 TFs and 1283 targets (Figure 6b; Appendix A). The most influential node in this GRN was TRITD5Bv1G096580, which belongs to the DOF TF family (Figure 6b). According to the Ensembl annotation, TRITD5Bv1G096580 has three orthologs in *T. aestivum* (TaDof2, TaDof3, and TaDof6) belonging to the same homoeologous group. A recent genome-wide analysis of the DOF family in wheat showed that these three ortholog genes belong to a separate cluster of the clade B and are clustered together with the rice RPBF gene (OsDof7) in the phylogenetic tree [96]. It was demonstrated in a previous study that the RPBF gene is preferentially expressed in maturing endosperm and can trans-activate several seed-specific genes implicated in storage protein and metabolic genes [97]. Interestingly, the ortholog genes in hexaploidy wheat (TaDof2, TaDof3, and TaDof6) were highly expressed in the early and middle stages of grain development, as previously reported in rice [96]. Moreover, it has been shown through transient expression assays in Arabidopsis protoplasts that TaDof2, TaDof3, and TaDof6 proteins are localized in the nucleus, which is consistent with their predicted molecular function [96].

In summary, multiple lines of evidence have supported the idea that the wheat homoeologous TaDof2, TaDof3, and TaDof6 are the functional orthologs to the rice RPBF gene and therefore could be important regulators of seed store reserve genes in wheat. Interestingly, we found that the corresponding orthologs in durum wheat (TRITD5Bv1G096580 and TRITD5Av1G116250) were downregulated by heat stress in developing grains according to the RNA-seq and qPCR analyses (Figure 7a and Appendix A). GO enrichment analysis of the TRITD5Bv1G096580 predicted targets showed that this TF is associated with glycogen and starch biosynthesis in response to HS (Figure 7b and Appendix A). Moreover, we scanned 500 bp upstream regions from the transcription start site for DOF binding sites in their predicted target genes, using the position weight matrices from the JASPAR 2020 database with CiiiDER software [54], and we found several significant enriched DOF binding sites (*p*-value < 0.05) in the promoter regions of more than 150 target genes (Figure 7c). Specifically, the most enriched motif was MA0984.1, which corresponded to the Arabidopsis DOF5.7 and was found in 424 out 438 target genes (Figure 7c). As shown in Figure 7d, three target genes encoding a key enzyme of starch biosynthesis in wheat such as ADP-glucose pyrophosphorylase [98] have between four and six DOF binding sites in their promoters; this suggested that the DOF TF might directly regulate the expression of genes involved in starch biosynthesis. The inhibitory effect on the expression of genes involved in carbohydrate metabolism in response to HS likely causes a general decrease in seed carbohydrate content as it was shown by the lower starch concentration of grains under HS (Figure 2e). 

## 4. Conclusions

In summary, our work focused on the characterization of the durum wheat grain response to high-temperature conditions during one of the key stages in the development of the grains and, at the same time, the identification of the molecular factors underlying this stress response. To this end, we used a strategy that integrated massive sequencing analysis and physiological characterization of grain weight and size, to elucidate some of the key mechanisms that govern this process.

We found that grain weight, size, and quality were severely on tetraploid wheat by short HS, in agreement with previous works on hexaploidy wheat [8,10,16,17,21,28,29,62]. During the early GF, the cell division and growth of the endosperm occur rapidly [8], at the same time several yield components such as length, width, and weight begin to be fixed in this stage; therefore, thermal increases would negatively affect the fixation of these trait yields. With respect to the grain quality, we observed opposite alterations in the levels of starch and proteins in the durum grain. The starch content was reduced because of the high temperatures and the protein concentration increased. Future analyses will be necessary to determine if this positive increase in protein content in durum wheat is accompanied by a change in the relationship between gliadins and glutenins, as previously described in bread wheat [17,62].

The initial transcriptomic response to HS in grain was characterized by the induction of chaperones, together with the inhibition of genes related to proteolysis and transcriptional regulation. This implicated that HS not only induced the expression of chaperones but also genes associated with photosynthesis, which has been proposed as a regulatory mechanism to induce acclimation and photoprotection during short-term HS [85]. On the other hand, the inhibitory effect of HS on genes involved in carbohydrate metabolism likely causes a general decrease in seed carbohydrate content. The TF-target gene regulatory network analysis identified a DOF TF (TRITD5Bv1G096580) gene as an important regulatory hub in the response to HS in durum wheat grains. Moreover, we found that the predicted targets of this TF are genes related to glycogen and starch biosynthesis, so this TF could be an important regulator of seed store reserve genes in durum wheat.

## Figures and Tables

**Figure 1 plants-11-00059-f001:**
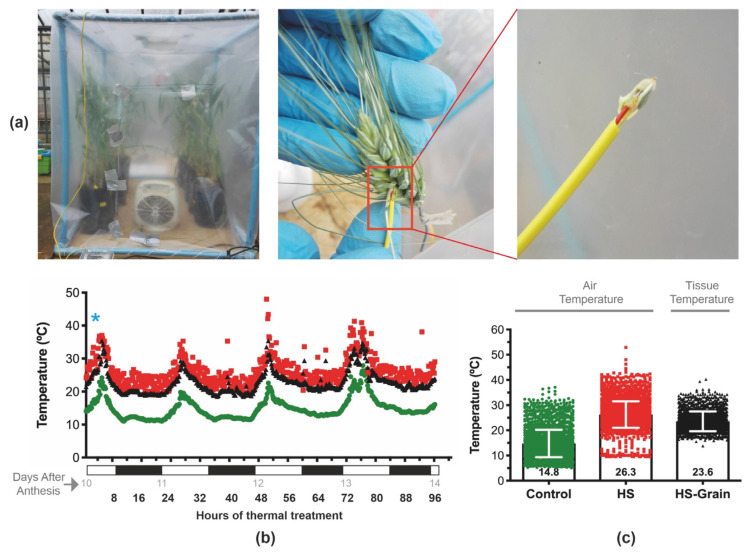
Heat stress chamber and temperature conditions of thermal treatments. (**a**) Wheat plants during heat treatments in polyethylene chambers (left), thermocouple in a wheat spike (middle), and a close-up of the probe inside a grain (right). (**b**) Temperature profiles from a representative experiment during 4 days of thermal treatments. Temperatures were recorded every 10 to 15 min. Control (glasshouse ambient air temperature in green dots), heat-stress (HS) treatment (air temperature inside of chambers in red squares), and tissue temperatures in grain during heat stress (black triangles). The blue asterisk shows the time when the transcriptomic analysis was performed. The alternating black and white bands on the X-axes indicate the night and day hours, respectively. The number of “Days After Anthesis” is indicated below X-axis in gray color. (**c**) Graphical display of temperature data from five independent thermal experiments. Inside each column, mean and error bar. The tissue temperatures in grain were recorded only in three independent experiments.

**Figure 2 plants-11-00059-f002:**
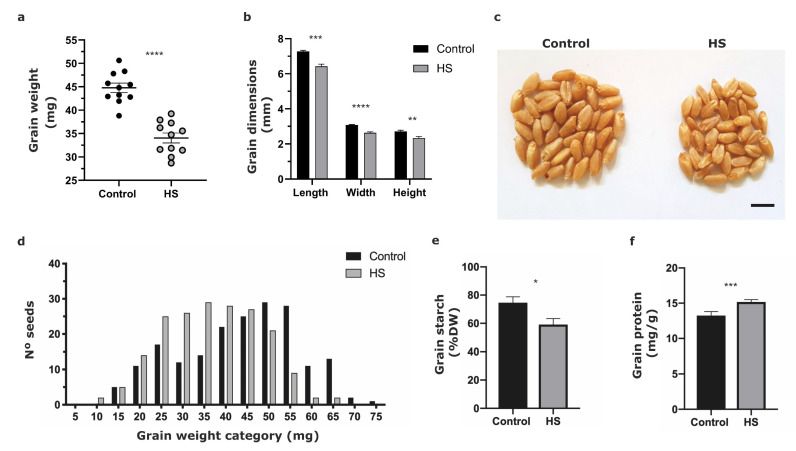
Physiological characterization of durum wheat grains at harvest under HS and control treatments. (**a**) Grain weight. (**b**) Grain dimensions. (**c**) Grain phenotype (40 seeds for each group, Bar: 0.5 cm). (**d**) Grain weight distribution. (**e**) Grain starch and (**f**) grain protein levels. Control (black columns) and heat treatment (gray columns). Each column represents the mean of three replicates with ±standard deviations. Significant effects at Student’s *t*-test, *p* < 0.05 (*), *p* < 0.01 (**), *p* < 0.001 (***), *p* < 0.0001 (****).

**Figure 3 plants-11-00059-f003:**
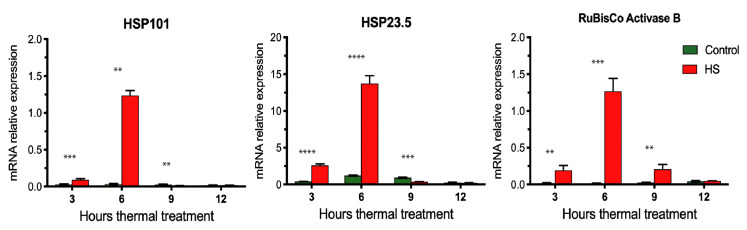
Temporal expression profiles of HS marker genes in durum wheat grains. Expression levels of three heat-stress gene markers at 3, 6, 9, and 12 h of starting thermal treatments by real-time qPCR. Heat-stress gene markers: heat shock protein (HSP) 101, HSP 23.5, and the Rubisco Activase B genes. Control: green color, and heat stress (HS): red color. Each column represents the mean of three biological replicates with ± standard deviations. Significant effects at Student’s *t*-test, *p* < 0.01 (**), *p* < 0.001 (***), *p* < 0.0001 (****).

**Figure 4 plants-11-00059-f004:**
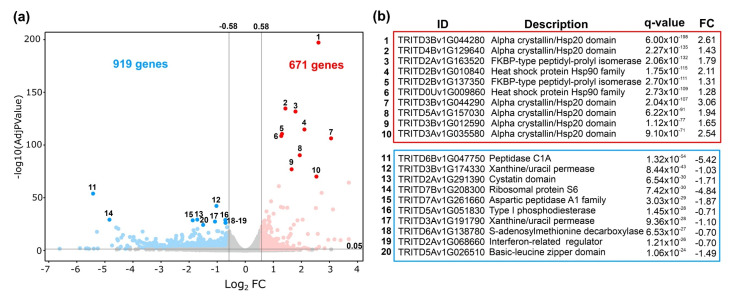
Volcano plot of differentially expressed genes identified between the heat-stress and control grain samples. (**a**) The blue dots denote downregulated gene expression (919 genes), the red dots denote upregulated gene expression (671 genes), and the gray dots represent genes under the threshold (*q*-value < 0.05 and absolute fold of change (FC) > 1.5). Numbers represent the top 10 of down- and upregulated genes. (**b**) The upper panel (red) shows the ten most significant upregulated genes. Lower panel (blue) the ten most significant downregulated genes. ID: Gene identification from *Triticum turgidum durum* (Svevo.v1). Description: Gene family. FC: log_2_ fold of change.

**Figure 5 plants-11-00059-f005:**
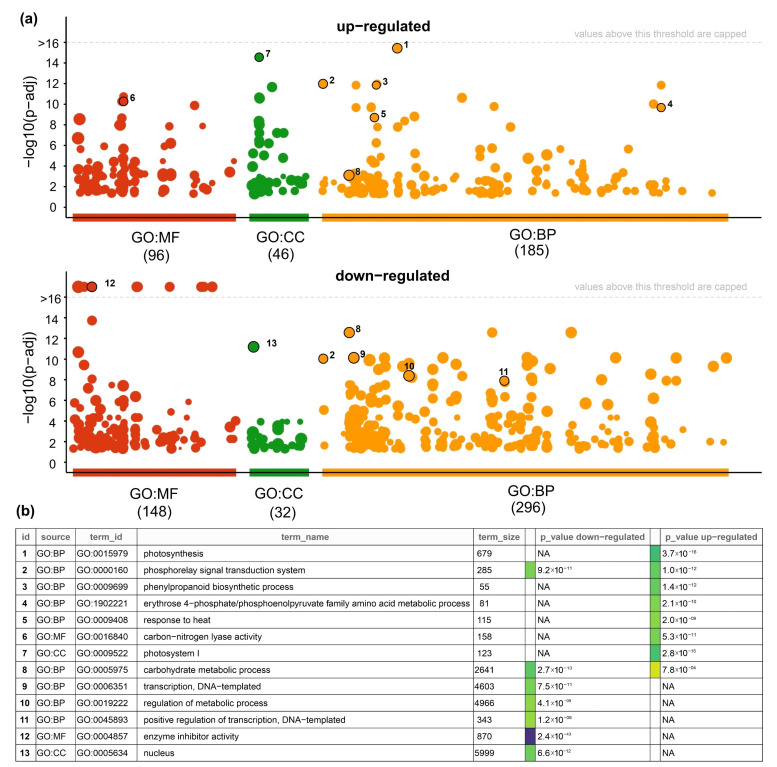
Gene ontology (GO) enrichment of differentially expressed genes (DEGs) based on transcriptomic data from short heat stress in durum grain wheat. (**a**) GO analysis of DEGs associated with a molecular function (MF) in red, cellular component (CC) in green, and biological process (BP) in orange. The upper panel represents the upregulated genes and the lower panel the downregulated genes. The numbers in parentheses denote the total numbers of GO terms for each category. (**b**) Table with a ranking of the five most significantly enriched biological processes for down- and upregulated genes. In addition, the most enriched GO term of the MF and CC domain is also indicated. The dot labels with a number (1 to 13) in (**a**) are described in (**b**). Each circle in the Manhattan plot represents a significant GO term (adjusted *p*-value < 0.05) using gprofiler2 software [51], and colors indicate the GO domain (MF: molecular function, CC: cellular component, BP: biological process).

**Figure 6 plants-11-00059-f006:**
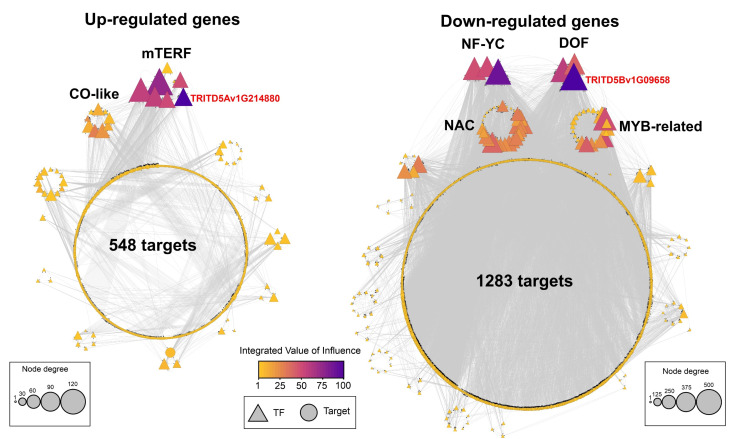
TF-target gene regulatory network based on DEGs from heat stress in durum grain wheat. Triangles represent TFs; circles represent targets. Gray lines denote the TF-target connections. TF, transcription factor. The color gradients from yellow to purple represent the integrated value of influence for each network [50].

**Figure 7 plants-11-00059-f007:**
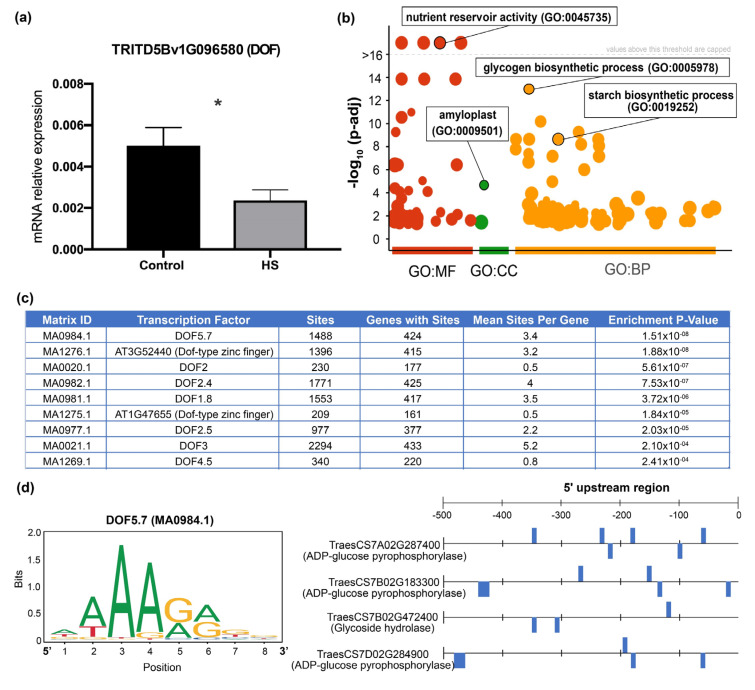
The predicted targets of a highly connected DOF TF gene (*TRITD5Bv1G096580*) are involved in glycogen and starch biosynthesis. (**a**) qPCR analysis of the *TRITD5Bv1G096580* gene in response to 3 h of HS in durum wheat grains. Relative expression with ± standard deviations. Significant effects at Student’s *t*-test, *p*-value < 0.05 (*). (**b**) GO enrichment analysis of 438 target genes of TRITD5Bv1G096580 DOF TF. Each circle in the Manhattan plot represents a significant GO term (adjusted *p-*value < 0.05) using gprofiler2 software [51], and colors indicate the GO domain (MF: molecular function, CC: cellular component, BP: biological process). (**c**) The 10 most enriched DOF motifs in the promoter region of 438 target genes of *TRITD5Bv1G096580*. The over-representation analysis was performed using the CiiiDER tool [54] and JASPAR 2020 database [53]. (**d**) Left panel: Base frequency analysis for the DOF5.7 (MA0984.1) motif obtained from the JASPAR 2020 database binding site [53], which is in the promoter sequences of putative target genes. Right panel: Schematic representation for the number and distribution of the DOF5.7 (MA0984.1) motif binding site (blue square) in gene promoter sequences of key enzymes of starch biosynthesis.

## Data Availability

The RNA-Seq datasets generated and analyzed during this study are available in the NCBI Gene Expression Omnibus (GEO) repository, accession GSE186472. All other data generated during this study are included in this published article and its Appendix A.

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
