# Peer review of "Transcriptomic and Physiological Response of Durum Wheat Grain to Short-Term Heat Stress during Early Grain Filling"

_plants, 2021, doi:10.3390/plants11010059_

Round 1

Reviewer 1 Report

The paper written by Arenas-M et al describes physiological response of durum wheat grain with the response to short-term heat stress during early grain filling. The results shows that a short-term heat-stress (HS) treatment leads to a significant reduction in grain weight and size, a decrease in starch content but the increments in grain protein level. The authors identified 1590 differentially expressed genes related to photosynthesis, heat response and carbohydrate metabolic process by using the transcriptomic data. Further, the authors discovered a novel TF involved in abiotic stress and grain quality.

Overall, the methods are thorough. Conclusions are appropriate and supported by the data. Data are available either within the manuscript, or at NCBI GEO. The manuscript is sound, and below I have highlighted a few suggestions for changes.

I suggest to accept the paper after the minor changes.

1 Line 338, few studies

2 Line 362, the dynamics of gene expression

3 Line 388, the authors should mention the differences or similarities between HSP101 and HSP23.5. It is better to understand whether they are playing same functions as HSP.

4 Line 423, q-value < 9.10e-71

5 Line 494, since the authors use q-value for the significance in the manuscript, it should be consistent here for the q-value instead of adjusted p-value

Author Response

Point-by-point response to comments:

Reviewer 1

I suggest to accept the paper after the minor changes.

RESPONSE: We appreciate the positive comment and suggested corrections by the reviewer.

 1 Line 338, few studies

RESPONSE: Done: “other studies have shown…”

2 Line 362, the dynamics of gene expression

RESPONSE: Thank you for noticing us this mistake, it has been corrected now.

3 Line 388, the authors should mention the differences or similarities between HSP101 and HSP23.5. It is better to understand whether they are playing same functions as HSP.

RESPONSE: Thank you for the suggestion. We clarified this raised point by adding the following sentence:

“In the case of HSPs, we selected HSP101 and HSP23.5 for expression analysis by qPCR (Figure 3), which belongs to the HSP100 and small HSPs families respectively. Members of both families have been shown that play a crucial role in thermotolerance acting as molecular chaperones protecting their targets from denaturation and aggregation [68–70]”

4 Line 423, q-value < 9.10e-71

RESPONSE: Thank you for noticing us about this mistake, it has been corrected in the revised version of the manuscript.

5 Line 494, since the authors use q-value for the significance in the manuscript, it should be consistent here for the q-value instead of adjusted p-value

RESPONSE: Thank you for your suggestion, done.

Reviewer 2 Report

This work focused on the characterization of the durum wheat grain response to high-temperature conditions during one of the key stages in the development of the grains and, at the same time, the identification of the molecular factors underlying this stress response.

The authors have done a great job. This study is fundamental and interesting, since the data obtained during the study can be used to create technologies to increase wheat grain production in conditions of global warming. The work was carried out at a high level using the most modern research methods, such as RNA-Seq analysis and bioinformatic analysis of data.

The article is well written, the introduction reflects the current state of the problem under study, the materials and methods are described in detail, the drawings are well presented and the results are described. The conclusion is concise and reflects all the results obtained.

I have no critical comments to the article. My comments are rather aimed at eliminating misprint and inaccuracies.

Decision: - Accept after minor revisions (which the authors can be trusted to make)

- Minor Revisions

1) line 175: uL perhaps it is better to replace with µL?

2) line 198, 200: (Applied Biological Materials Inc.)/(Agilent) country-manufacturer?

3) line 307 and further in the text: p-value (p < 0.01) should be italicized;

4) Fig 1. Conditions of thermal treatments and gene expression of heat-stress marker genes. Gene expression is not represented in this figure. The name of Fig.1 should be corrected;

5) line 365: “Heat shock proteins (HSPs are crucial…” missing the round bracket after HSPs.

Author Response

Thank you very much for reviewing our manuscript and the overall positive comments.

1) line 175: uL perhaps it is better to replace with µL?

RESPONSE: Thank you, this mistake has been fixed.

2) line 198, 200: (Applied Biological Materials Inc.)/(Agilent) country-manufacturer?

RESPONSE: Thank you for this observation, the country-manufacturer has been included in the new version of the manuscript.

3) line 307 and further in the text: p-value (p < 0.01) should be italicized;

RESPONSE: Thank you for highlighting this mistake, it was fixed.

4) Fig 1. Conditions of thermal treatments and gene expression of heat-stress marker genes. Gene expression is not represented in this figure. The name of Fig.1 should be corrected;

RESPONSE: Thank you for pointing out this mistake, it was fixed.

5) line 365: “Heat shock proteins (HSPs are crucial…” missing the round bracket after HSPs.

RESPONSE: Thank you for noticing this mistake, done.

Reviewer 3 Report

The manuscript deals with the "wheat responses to the short-term heat stress". In overall, manuscript has been written well.

Wheat (Triticum aestivum L.) is the third most vital cereal in the world after rice and maize. Therefore, wheat was selected to discuss cadmium uptake by plants in the current work. Almost 60% of the wheat produced globally is consumed as food, and wheat demand is globally expected to rise by an estimated 70% in the next few decades (2020–2050) as the human population increases and rising income levels increase household consumption https://doi.org/10.3390/plants9040500  

1. The main question here is about the novelty of the study. Many studies have been already conducted about the effects of different types of stresses (including heat stress) on wheat.

https://doi.org/10.1007/s13593-017-0443-9

https://doi.org/10.1371/journal.pone.0222639

https://doi.org/10.3389/fpls.2021.739246

https://www.iomcworld.org/articles/heat-stress-effects-and-tolerance-in-wheat-a-review-53182.html

2. Write keywords alphabetically.

3. Page 1, Line 17; "We found a significant reduction in grain weight and size from HS treatment."

Mention the values.

4. Page 1, Line 18; ".. showing a decrease in starch content in addition to increments in grain protein levels."

Same comments as above. Mention the values.

5. Page 2, Line 51; "These studies have suggested that the wheat sensitivity to HS largely depends on the timing and intensity of the stress"

If the effects of HS on wheat is well-known; thus, what is the novelty of your study. Try to highlight the lack of previous studies in the Introduction.

Author Response

Thank you very much for reviewing our manuscript and the overall positive comments.

  1. The main question here is about the novelty of the study. Many studies have been already conducted about the effects of different types of stresses (including heat stress) on wheat.

RESPONSE: We agree that several heat-stress studies have been carried out on wheat. In fact, we included the suggested references by the reviewer as well as other (e.g. Kino et al., 2020) in this revised version of the manuscript. However, these references and previous heat-stress studies focused on bread wheat looking for the heat effect on different tissues (e.g. flag leaf) and phenological timings of our work (see lines 69-70). In contrast to previous studies, we focused on analyzing the response to heat stress during few days (4 days) at the beginning of grain growth (10DAA) when the grain is still enlarging but the linear dry matter accumulation is beginning, which is critical for grain weight determination (Herrera and Calderini, 2020). Therefore, the timing of our experiment is different from several other studies. For example, Kino et al. (2020) imposed the heat increase from 6 to 18 DAA. Additionally, the transcriptome evaluation was carried out 0, 4 and 8 days after setting the heat treatment. According with the previous studies, little is known about the early transcriptome response during grain filling and we did not find a similar work in Triticum durum. As a consequence, this is the first study that analyze the initial transcriptomic response to short-term heat stress (3 hours after setting the heat treatment) in durum wheat grains. Another novel contribution of our manuscript is the identification of key transcription factors (TFs) related to HS response in grains. We integrated TF-target interaction data with differentially expressed genes to construct gene regulatory networks. These networks were analyzed with state-of-the-art algorithms such as IVI (https://doi.org/10.1016/j.patter.2020.100052), which allowed us to identify novel TFs have not yet been characterized in the framework of HS responses in grains such as mTERF, CO-like, DOF, NF-YC or NAC. Functional studies of these new candidate genes should improve the understanding of the regulatory mechanisms underlying heat stress responses in grain durum wheat.

  1. Write keywords alphabetically.

RESPONSE: Thank you for the suggestion, we arranged the keywords as suggested.

  1. Page 1, Line 17; "We found a significant reduction in grain weight and size from HS treatment."

Mention the values.

RESPONSE: Thank you, the values were added in this revised version of the manuscript. 

  1. Page 1, Line 18; ".. showing a decrease in starch content in addition to increments in grain protein levels."

Same comments as above. Mention the values.

RESPONSE: as the previous suggestion, the values were included in this revised version of the manuscript. 

  1. Page 2, Line 51; "These studies have suggested that the wheat sensitivity to HS largely depends on the timing and intensity of the stress"

If the effects of HS on wheat is well-known; thus, what is the novelty of your study. Try to highlight the lack of previous studies in the Introduction.

RESPONSE: As it was stated above, our study provides novel information on the response of durum wheat to heat stress. The agreement of our study, and many other previous ones, on that HS largely depends on the timing and intensity of the stress does not mean lack of originality. Actually, many papers concur on general crop responses. The novelty of our manuscript is not due to the statement of the importance of timing and intensity, the originality is supported by the timing of our heat treatment, the early transcriptome evaluation, the experiment, the identification of key transcription factors (TFs) and the lack of knowledge on the response of durum wheat to heat stress. Anyway, we accepted the reviewer suggestion and included the following sentences in the introduction section, highlighting the novelty of our study:

“However, the initial transcriptomic response of grains to HS during early GF has not yet been analyzed in durum wheat. Moreover, the molecular factors controlling the early events of the HS response of durum wheat grains remain unknown.”

Round 2

Reviewer 3 Report

No further comments!